# Reduced FXR Target Gene Expression in Copper-Laden Livers of COMMD1-Deficient Dogs

**DOI:** 10.3390/vetsci6040078

**Published:** 2019-09-30

**Authors:** Xiaoyan Wu, Hsiaotzu Chien, Monique E. van Wolferen, Hedwig S. Kruitwagen, Loes A. Oosterhoff, Louis C. Penning

**Affiliations:** Department Clinical Sciences of Companion Animals, Faculty of Veterinary Medicine, Utrecht University, P.O. BOX 80.154, NL-3508 TD Utrecht, the Netherlands; X.Wu@uu.nl (X.W.); vltchien11@gmail.com (H.C.); M.E.vanWolferen@uu.nl (M.E.v.W.); H.S.Kruitwagen@uu.nl (H.S.K.); L.A.Oosterhoff@uu.nl (L.A.O.)

**Keywords:** copper toxicosis, Bedlington terrier, nuclear receptor, COMMD1

## Abstract

Wilson’s disease (WD), an autosomal recessive disorder, results in copper accumulation in the liver as a consequence of mutations in the gene ATPase copper transporting beta (*ATP7B*). The disease is characterized by chronic hepatitis, eventually resulting in liver cirrhosis. Recent studies have shown that dysregulation of nuclear receptors (NR) by high hepatic copper levels is an important event in the pathogenesis of liver disease in WD. Intracellular trafficking of ATP7B is mediated by COMMD1 and, in Bedlington terriers, a mutation in the *COMMD1* gene results in high hepatic copper levels. Here, we demonstrate a reduced Farnesoid X nuclear receptor (FXR)-activity in liver biopsies of *COMMD1*-deficient dogs with copper toxicosis, a unique large animal model of WD. FXR-induced target genes, small heterodimer partner (SHP), and apolipoprotein E (ApoE) were down-regulated in liver samples from *COMMD1*-deficient dogs with hepatic copper accumulation. In contrast, the relative mRNA levels of the two CYP-enzymes (reduced by FXR activity) was similar in both groups. These data are in line with the previously observed reduced FXR activity in livers of *ATP7B−/−* mice and WD patients. Therefore, these data further corroborate on the importance of the *COMMD1-*deficient dogs as a large animal model for WD.

## 1. Introduction

Wilson’s Disease (WD) is an inherited disease characterized by excessive hepatic accumulation of the trace element copper, leading to chronic liver disease and cirrhosis [1]. This autosomal recessive disorder is caused by mutations in the gene coding for *ATPase copper transporting beta* (ATP7B) [2]. Mutations in *ATP7B* result in impaired biliary copper excretion and, subsequently, in the accumulation of copper in tissues, especially in the liver. Copper overload may lead to Fenton-type redox reactions, oxidative stress, and cellular damage, which are correlated with the progression and pathology of the disease [3,4]. However, it seems unlikely that copper-induced Fenton-like reactions are indeed causative in copper overload models [5,6]. Therefore, another explanation of copper overload-induced cellular disturbances needs to be investigated.

In 2015, Wooton-Kee et al. revealed that elevated hepatic copper levels reduced nuclear receptor (NR) function in WD patients [7]. NRs work as transcriptional factors by direct binding (as homodimers or heterodimers) to specific sequences of response elements on the DNA, thereby up- or down-regulating the transcription of specific gene products. The copper-mediated reduced function of the NRs farnesoid X receptor (FXR) and liver X receptor (LXR) seem to play an important role in development of liver pathology in WD [8]. Whereas WD and idiopathic forms of copper-associated hepatitis are rare diseases in humans, copper-associated hepatitis is often encountered in dogs. In two dog breeds, the mutations causing copper toxicosis have been elucidated; namely, a deletion of exon-2 of the *COMMD1* in Bedlington terriers [9], and both causative (*ATP7B:c.4358G>A*) and modifier (*ATP7A:c.980C>T*) mutations in Labrador retrievers [10]. Recently, we have observed a significant association of the *ATP7B:c.4358G>A* mutation and elevated hepatic copper concentrations in combined Dutch and American Dobermans [11].

We investigated an inherited copper toxicosis, other than *ATP7B*-mediated, in order to see whether the effects on nuclear receptors were specific for ATP7B or were more generally caused by increased hepatic copper concentrations. This COMMD1-deficient model resembles WD at histology [12,13], even though it is different from WD and *ATP7B−/−* mice, with respect to the genetic culprit. Hence, we analyzed hepatic NR activity indirectly, by FXR target gene expression in copper–laden canine livers from homozygous *COMMD1*-deficient dogs, and non-copper-laden livers from heterozygous and wildtype dogs [13].

## 2. Materials and Methods

### 2.1. Ethics, Animals, and Samples

Liver samples from wildtype, heterozygous, and homozygous *COMMD1*-deficient dogs were obtained, as previously described [12]. The absence or presence of liver disease was confirmed histologically by a board-certified veterinary pathologist and the elevated hepatic copper levels of the *COMMD1*-deficient dogs have already been described [12]. Data were collected according to the Act on Veterinary Practice and the procedures were approved by the local ethics committee, as required under Dutch legislation (ID 2007.III.08.110).

### 2.2. Genotyping and Phenotyping

The COMMD1 status of the Beagle/Bedlington terriers and the hepatic copper status has been described previously [12]. These longitudinal studies clearly showed a time-dependent increase in hepatic copper accumulation, from base line (less than 400 parts per million (ppm) in healthy animals towards greater than 4000 ppm for COMMD1-deficient dogs), development and progression of hepatitis, and deposition of extracellular matrix (fibrosis).

### 2.3. RNA Isolation, cDNA Synthesis, and q-RT-PCR

RNA was isolated with Qiagen columns and included an on-column DNase-I treatment to minimize genomic DNA contaminations. Relative gene expression of the selected genes—small heterodimer partner (*SHP*), Apolipoprotein E (*ApoE*), cholesterol 7 alpha-hydroxylase (*CYP7A1*), and sterol 12 alpha-hydroxylase (*CYP8B1*)—was measured using RT-qPCR on cDNA obtained using the iScript™ cDNA synthesis kit as described by the manufacturer (Bio-Rad, Veenendaal, the Netherlands). A mix of random hexamers and oligo-dT primers was used to secure optimal cDNA synthesis. Primer design, validation, RT-qPCR conditions, and data analysis were as described previously [14]. Briefly, Perlprimer v1.1.14 was used for primer design with Ensembl annotated transcripts and MFold was used to test for secondary structures in the amplicons [15]. Primer specificity was obtained in silico (BLAST analysis) and by sequencing; qPCR was performed using primers as described in Table 1, and was based on the high affinity double-stranded (ds) DNA-binding dye SYBR green I (iQSYBR Green Supermix, BioRad Veenendaal, the Netherlands) [16]. As the fluorescence of SYBR green I is sequence-independent, a melt curve analysis was performed for each sample after the RT-qPCR, in order to confirm the amplification of one single amplicon. For quantification, normalization was performed using the geometric mean of four reference-genes: hypoxanthine phosphoribosyltransferase (HPRT), hydroxymethylbilane synthase (HMBS), signal receptor particle receptor (SRPR), and ribosomal protein S5 (RPS5) [17]. Expression stability of these four reference genes was analyzed, and this combination revealed a stable expression, according to the GeNorm algorithm [18]. Stable expression of reference genes is an important and crucial factor in the normalization procedure, in order to quantify the relative expression of gene products, as required under the minimum information for publication of quantitative real-time PCR experiments (MIQE) precise guidelines [19].

### 2.4. Isolation of Canine Biliary Duct Fragments and Liver Organoid Culture

Hepatic progenitor cells were isolated from canine liver tissue upon mechanical dissection and subsequent type-II collagenase (Gibco, Landsmeer, the Netherlands) and dispase (Gibco, Landsmeer, the Netherlands) digestion, as described previously [20]. Expansion medium (EM) conditions, medium changes and split ratio were as described, in detail, in [20]. Organoid stem cell differentiation towards hepatocytes was induced by the addition of 25 ng/mL BMP7 (Peprotech, Heerhugowaard, the Netherlands) after the last passage in expansion medium. Four days after the last passage, Wnt-conditioned medium, ROCK inhibitor, and Noggin were withdrawn from the medium. Further differentiation was acquired by withdrawal of nicotinamide, R-spondin-1-conditioned medium, and FGF10 six days after the last passage in expansion medium. Finally, BMP7 was continued and 100 ng/mL FGF19 (R&D Systems, Abingdon, UK), 10 µM DAPT (Selleckchem, Huissen, the Netherlands), and 30 µM dexamethasone (Sigma-Aldrich, Zwijndrecht, the Netherlands) were added. This culture was continued for over one week. The lentiviral transduction to supplement *COMMD1-*deficient organoids with the correct COMMD1 coding sequence was as described previously [20].

### 2.5. Statistical Analysis

Data are presented as the medians and range. Data were non-normally distributed (Shapiro-Wilk test) and, therefore, a Kruskal–Wallis test was performed to detect any significant differences among multiple groups. Then, a Mann–Whitney U test was used to compare differences between the two groups. A *p* ≤ 0.05 was considered significantly different. All statistical analyses were performed with RStudio (Version 1.1.383—© 2009–2017 RStudio, Inc., Boston, USA) [21].

## 3. Results

The hepatic copper levels were severely increased (>4000 ppm) only in the *COMMD1*-deficient dogs, as described previously for these samples [12]. Both WT and heterozygotes were clinically healthy and had non-elevated hepatic copper levels [12]. To measure FXR activity, we analyzed the relative expression of four gene products which are induced by FXR; namely, *SHP*, *Bile Salt Export Pump* (*BSEP*), *Ileal Bile Acid Binding protein gene* (*IBABP*), and *APOE.* In contrast, *CYP7A1* and *CYP8B1* transcription is suppressed by FXR activity.

As no difference in relative mRNA expression levels of the six measured FXR target genes was observed between *COMMD1+/+* and *COMMD1+/−* liver samples (data not shown)—as expected, based on the normal hepatic copper levels and healthy histology—these two groups were combined and compared to samples from copper-laden livers of *COMMD1*-deficient dogs. As depicted in Figure 1, the relative mRNA expression of *SHP* was reduced about 3-fold in *COMMD1*-deficient livers (*COMMD1−/−*), as compared to clinically heathy samples (*p* = 0.03; Figure 1A). A small, but significant (*p* = 0.029), reduction in *APOE* mRNA expression in the copper-laden *COMMD1−/−* liver samples was measured (Figure 1B). Levels of *IBABP* were very low, as expected from a gene product mainly expressed in the intestine, and even undetectable in organoids. Although the relative expression was reduced, in line with FXR activity, in the *COMMD1*-deficient livers, the very low expression levels did not justify a significance claim and we did not further elaborate on this gene product (data not shown). Similarly, relative mRNA levels of *BSEP* were below detection level in organoids; whereas, in the whole livers, relative expression levels were not significantly different (data not shown). In contrast to the genes that are upregulated by FXR activity, the relative mRNA levels of the two CYP-enzymes (which are reduced by FXR activity) were similar in both groups (Figure 1C,D).

### 3.1. Relative mRNA Expression of SHP, APOE, CYP7A1, and CYP8B1 in Canine Liver Samples

### 3.2. Relative mRNA Expression of COMMD1 +/+ Organoids in Expansion Medium (EM) and Differentiation Medium (DM)

As liver organoids are described as model systems for nuclear receptor studies in mice, we extended our studies with canine liver organoids [20,22]. Canine liver organoids (*COMMD1+/+*) cultured in EM had undetectable *APOE* expression levels, whereas the relative expression of *SHP*, *CYP7A1*, and *CYP8B1* was lower, compared to the WT liver expression (Figure 1 and Figure 2), especially for the biotransformation gene products *CYP7A1* and *CYP8B1*. Modifications of the culture medium, aiming to differentiate the organoids into a hepatocyte-like phenotype (DM, differentiation medium) resulted in relative *SHP* levels comparable to the levels observed for WT livers (Figure 1A and Figure 2A). The relative mRNA expression of these three gene products measured in DM was not significantly increased, compared to organoids cultured in EM (Figure 2**A**–**C**). Interestingly, the relative expression of *SHP* in DM organoids was similar to that for total liver samples. In contrast, the relative expression of the two biotransformation gene products remained 30–100-fold lower in organoids cultured in DM, compared to the relative expression in total liver samples (Figure 2B/C versus Figure 1C/D).

### 3.3. Relative mRNA Expression Levels in Organoids of COMMD1+/+, COMMD1−/−, and COMMD1−/−Transduced

As the relative expression of *SHP*, *CYP7A1*, and *CYP8B1* was detectable in organoids differentiated towards hepatocytes, we continued to investigate their relative mRNA expression levels in organoids of the following genotypes: *COMMD1+/+*, *COMMD1−/−*, and *COMMD1−/−* supplemented with the full *COMMD1* gene (Figure 3). The canine hepatic organoids were cultured without additional copper, in order to investigate a direct COMMD1 effect on FXR activity. No significant differences were observed between the three groups, except that *CYP8B1* relative mRNA expression was significantly reduced in supplemented *COMMD1*-deficient organoids, compared to *COMMD1+/+* (*p* = 0.016; Figure 3C).

## 4. Discussion

In order to gain a better insight into the link between hepatic copper accumulation, COMMD1 protein expression and nuclear receptor (FXR) activity, we investigated the activity of the nuclear receptor FXR [23] in both *COMMD1*-deficient canine liver tissue, as well as canine liver organoids. Organoids have been shown to be a very good in vitro model for studying nuclear receptor biology [22]. FXR is highly expressed in the liver and is involved in the regulation of bile acid, cholesterol, triglyceride, and glucose metabolism [24]. By measuring the relative mRNA expression level of FXR target genes, we demonstrated that small heterodimer partner (*SHP*) and *APOE* have significantly lower mRNA expression levels in *COMMD1-*deficient dog livers than in non-affected livers. The relative expression levels of *CYP7A1* and *CYP8B1* point towards the same direction; namely, reduced FXR activity in copper-laden livers and/or cells. The fact that, regardless of low expression levels, *BSEP* expression was not different in the cases and controls, serves as conformation that indirect measurement of FXR activity through the expression of FXR target genes, as we did, does not fully represent the complexity of FXR activation. The complexity, for instance, lies in the fact that FXR can homodimerize and heterodimerize. Despite these obvious limitations to our study design, our data corroborates the findings by the Lutsenko group [7,8] on WD patients and *ATP7B−/−* mice. The murine, canine (liver tissue and organoids), and human data collectively implicate intrahepatic copper levels as a regulator of FXR activity. The canine hepatic organoids investigated in this assay were cultured without additional copper. Copper levels can be increased under culture conditions with excessive copper, as described before [20]. However, we were reluctant to do further investigations into this copper-overload model. By no means were the copper levels in the organoids in line with the excessive copper levels observed in livers of COMMD1-deficient dogs [12,14]. In this respect, it should be noted that the organoid copper levels were, at most, only doubled (with 5000 relative fluorescent units for WT and supplemented, and 10,000 for COMMD1-deficient organoids) in the *COMMD1*-deficient liver organoids cultured for 24 h in the presence of excessive copper [20]. Unfortunately, the amount of cells limited true quantitative copper measurements. Therefore, we did not further analyze FXR target gene expression under high copper culture conditions. This does not contradict the effects of COMMD1 on FXR activity, but merely indicates the importance of elevated copper levels as mediators of reduced FXR activity, rather than a COMMD1-mediated direct effect.

The limitation of this study lies in the way that the FXR activity was analyzed. FXR activity was analyzed indirectly, by means of expression of FXR target genes. Obviously, direct binding and activation of FXR by chromatin immunoprecipitation assays (ChIP-assays) is a preferred method of choice, able to more directly prove aberrant FXR activity. However, this relies fully on the availability of specific antibodies to precipitate FXR proteins. The lack of knowledge and availability of antibodies for canine FXR hampers such experiments. This technical limitation impedes FXR activity analysis in a direct fashion and, therefore, indirect assays (such as those described here) are the second-best option.

In livers with high copper levels (e.g., *ATP7B−/−* mice and WD and progressive familial cholestasis (PFIC) patients), the activity of FXR is reduced [7,8]. This finding shows that, other than *ATP7B*-related mutations causing elevated hepatic copper, a deletion of exon-2 of the *COMMD1* gene might yield a similarly reduced liver FXR activity. Together, these results agree with previous studies on *ATP7B* mutations, WD, and FXR activity [5,6]. To really quantify an association of hepatic copper with FXR activity, we need large cohorts of samples with a broad range of hepatic copper levels and an analysis of FXR activity in these samples. For instance, the Labrador retriever samples previously described can provide us with such a quantitative correlation [25]. This quantitative copper study has also been observed recently in Dobermans [11]. Hepatic copper overload is caused by mutations in the *ATP7B* gene and, in contrast to WD (which is a rare disease in humans), copper toxicosis in Labrador retrievers is a frequently observed clinical presentation [26].

## 5. Conclusions

In summary, these data not only confirm the previous data in other mammals, but further emphasize the importance of COMMD1-deficient dogs as a large animal model for WD [12,13].

## Figures and Tables

**Figure 1 vetsci-06-00078-f001:**
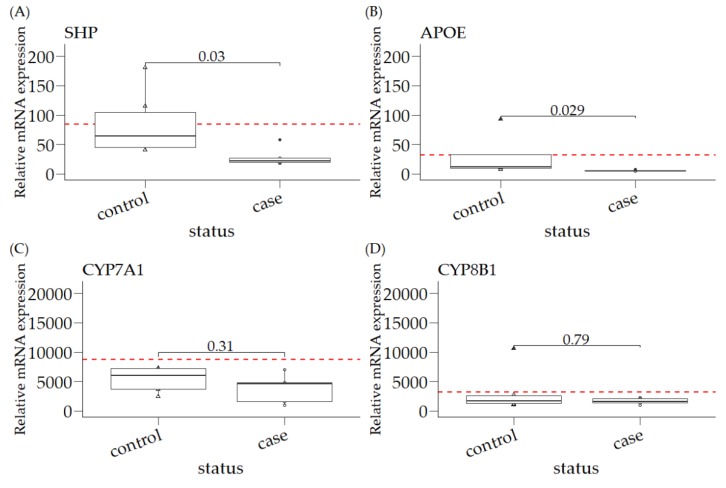
Relative mRNA expression of *SHP, APOE, CYP7A1,* and *CYP8B1* in canine liver samples. Controls are the combination of *COMMD1+/+ and COMMD1+/−* samples, which have normal hepatic copper levels (n = 6). Cases are *COMMD1−/−* deficient liver samples. Values are shown as the medians, interquartile range (IQR), and the minimum to maximum range. Statistical significance between case and control was assessed by a Mann–Whitney U test (*p* ≤ 0.05). (**A**)relative mRNA expression of SHP is significantly different between controls and cases (*p*-value = 0.03); (**B**) relative mRNA expression of APOE is significantly different between controls and cases (*p*-value = 0.029); relative mRNA expression is not significantly different between controls and cases for (**C**) CYP7A1 or (**D**) CYP8B1. The red dotted horizontal line is the mean value for the control group, in order to compare with the organoids cultured in differentiation medium (DM) in Figure 2.

**Figure 2 vetsci-06-00078-f002:**
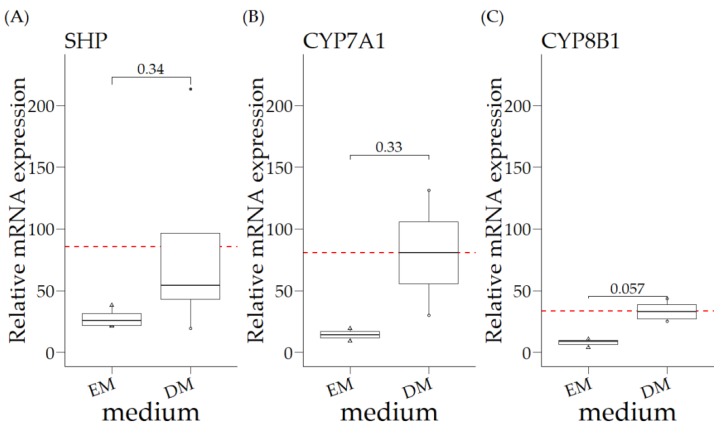
Relative mRNA expression of *COMMD1+/+* organoids in expansion medium (EM) and differentiation medium (DM). Statistical significance between EM and DM was assessed by a Mann–Whitney U test (*p* ≤ 0.05). Relative mRNA expression is not significantly different between different mediums for (**A**) SHP, (**B**) CYP7A1 or (**C**) CYP8B1. The red dotted line is the mean value of DM group, for comparison with the control liver tissues in Figure 1.

**Figure 3 vetsci-06-00078-f003:**
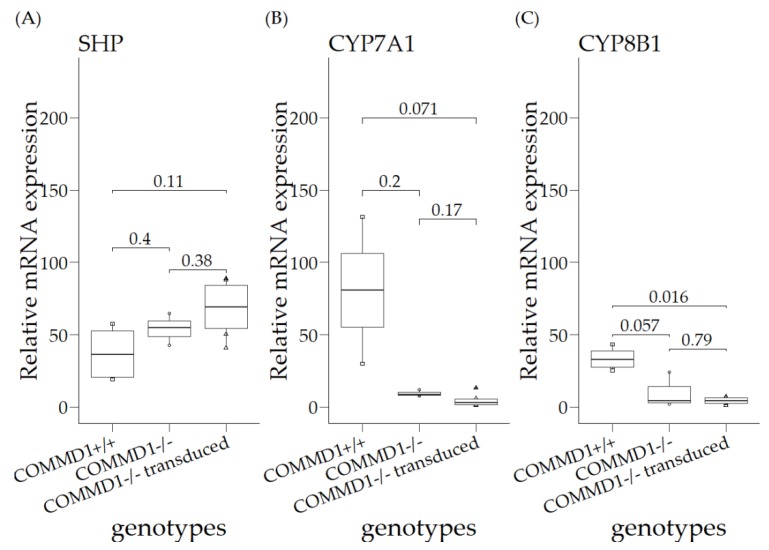
Relative mRNA expression levels in organoids of *COMMD1+/+*, *COMMD1−/−,* and COMMD1*−/−* transduced (*COMMD1−/−* supplemented with the *COMMD1+/+* gene). (**A**) SHP; (**B**) CYP7A1; (**C**) CYP8B1 Statistical significance among different genotypes was assessed by a Kruskal–Wallis test (*p* ≤ 0.05).

**Table 1 vetsci-06-00078-t001:** primer information for qPCR expression.

Genes	Sequences (5’ to 3’)	Annealing Temperature	Amplicon Size in bp
SHP	FW: AACATTCTCCCGTTTGACCAC	62 °C	99
	RV: GTAGTTGGCGTTGATGTAGTCG		
APOE	FW: CGCTTCTGGGATTACCTG	58 °C	124
	RV: CCTTCACCTCCTTCATGG		
CYP7A1	FW: TTTCAAATGATTAGGAGCCCTG	66 °C	115
	RV: TGATTCAGACAAATAGGACTGC		
CYP8B1	FW: CCAAGCATGGAGATGTGTTCAC	66 °C	162
	RV: CCCGAGTGGTATCCAAATACC		
HPRT	FW: AGCTTGCTGGTGAAAAGGAC	58 °C	104
	RV: TTATAGTCAAGGGCATATCC		
HMBS	FW: TCACCATCGGAGCCATCT	61 °C	112
	RV: GTTCCCACCACGCTCTTCT		
SRPR	FW: GCTTCAGGATCTGGACTGC	61 °C	81
	RV: GTTCCCTTGGTAGCACTGG		
RPS5	FW: TCACTGGTGAGAACCCCCT	63 °C	141
	RV: CCTGATTCACACGGCGTAG		

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
