# Peer review of "Reduced FXR Target Gene Expression in Copper-Laden Livers of COMMD1-Deficient Dogs"

_vetsci, 2019, doi:10.3390/vetsci6040078_

Round 1

Reviewer 1 Report

1.  Overall, this paper needs to be edited for clarity.  The clarity of the text is ok but really should be improved before the final publication.

2.  Line numbers 39/40 are do not adequately describe nuclear receptor function.  You say "NRs work as transcriptional factors by binding to response elements on DNA promoter sequences....." rather than (in)direct binding.  Some NRs (SHP, for example) regulate gene expression through indirect mechanisms, so if you want to describe that, you need to be clear in your statement.

3.  In the introduction, please state the reason for studying nuclear receptor activity in the COMMD model.  Something similar to the author's response in the cover letter would be sufficient.

4.  Although others have quantitated copper in these models, this paper would be significantly  improved if the authors either (a) measured copper in the samples used for their studies or (b) clearly showed previously published values in a table/figure - but, please make sure that this is cited, so the readers know where the data is coming from.  Given the novelty of the organoid culture in the COMMD model, it would be much better if the authors verified copper concentrations in their samples.

Reviewer 2 Report

The authors made correction and manuscript looks good.

Author Response

The authors made correction and manuscript looks good.

Thank you.

This manuscript is a resubmission of an earlier submission. The following is a list of the peer review reports and author responses from that submission.

Round 1

Reviewer 1 Report

The authors made correction and changes to the previous submission. I just have one comment on the figures. I do not thing that there is any significant differences in FIg 1,2 . The p value is high. Does * represent by p<0.05 ? 

Reviewer 2 Report

Overall, this is an interesting study.  However, the relevance of studying these pathways in canines should be elaborated in the introduction, i.e., why study nuclear receptors in a canine model  - is it different from the mouse/more similar to Wilson's disease patients.  The findings in the organoid cultures are unclear, since the FXR target gene (SHP) is not decreased.  This is explained in the discussion; however, showing representative copper data alongside expression would be helpful to the reader.   It would be more clear if the authors presented copper levels in the whole liver and organoid cultures to more clearly demonstrated the link between copper and nuclear receptor target gene expression. Further, ChIP assays have been done in mouse studies (FXR), so antibodies are available.  Is the issue that there aren't good FXR antibodies available for canine studies?  In lieu of ChIP, the authors can perform EMSA studies with the FXRE from target genes to study binding.  EMSA studies would strengthen the data presented.